

# Two sugar beet chitinase genes, *BvSP2* and *BvSE2*, analysed with SNP Amplifluor-like markers, are highly expressed after Fusarium root rot inoculations and field susceptibility trial

Raushan Yerzhebayeva[1], Alfiya Abekova[1], Kerimkul Konysbekov[2], Sholpan Bastaubayeva[1], Aynur Kabdrakhmanova[3], Aiman Absattarova[4] and Yuri Shavrukov[5]

[1] Kazakh Research Institute of Agriculture and Plant Growing, Almalybak, Almaty District, Kazakhstan
[2] Taldykorgan Branch, Kazakh Research Institute of Agriculture and Plant Growing, Taldykorgan, Almaty District, Kazakhstan
[3] I. Zhansugurov Zhetysu State University, Taldykorgan, Almaty District, Kazakhstan
[4] National Agricultural Research Education Centre, Astana, Kazakhstan
[5] College of Science and Engineering, School of Biological Sciences, Flinders University of South Australia, Bedford Park, SA, Australia

Corresponding author
Yuri Shavrukov,
yuri.shavrukov@flinders.edu.au

## ABSTRACT

**Background:** The pathogens from *Fusarium* species can cause Fusarium root rot (RR) and other diseases in plant species including sugar beet (*Beta vulgaris* L.), and they have a strong negative impact on sugar beet yield and quality.

**Methods:** A total of 22 sugar beet breeding lines were evaluated for the symptoms of RR after inoculation with *Fusarium oxysporum* Sch., isolate No. 5, and growth in a field trial. Two candidate genes for RR resistance, *BvSP2* and *BvSE2*, encoding chitinases Class IV and III, respectively, were previously identified in sugar beet, and used for genotyping using modern Amplifluor-like single nucleotide polymorphism (SNP) genotyping approach. The qPCR expression analysis was used to verify responses of the candidate genes for RR infections.

**Results:** A strong association of two SNP markers for *BvSP2* and *BvSE2* with resistance to RR in sugar beet was found in our study. Very high *BvSP2* expression (100-fold compared to Controls) was observed in three RR resistant accessions (2182, 2236 and KWS2320) 14 days after inoculation which returned to the control level on Day 18. RR sensitive breeding line 2210 showed a delay in mRNA level, reaching maximal expression of *BvSP2* 18 days after inoculation. The gene *BvSE2*, showed a strong expression level in leaf samples from the infected field trial only in the breeding line 2236, which showed symptoms of RR, and this may be a response to other strains of *F. oxysporum*.

## INTRODUCTION

Root rot (RR) has a strong negative impact on many crops, including sugar beet (*Beta vulgaris* L.), where both the quantity and quality of production from farmers can be significantly reduced (*Martin, 2003*; *Heydari & Pessarakli, 2010*; *Chehri, 2011*). The most important component to consider in the broad scale management of RR is host plant resistance and disease severity, where a greater investment must be paid to comparing the immune response of the available sugar beet breeding lines and varieties to RR pathogens (*Eggestein, Grazhdankin & Ugarov, 2008*; *Christ & Varrelmann, 2011*).

Phytopathological studies have revealed that a number of *Fusarium* species can cause Fusarium RR and other diseases in sugar beet like storage rot (*Lupashku & Mereniuc, 2010*; *Christ & Varrelmann, 2011*; *Liebe & Varrelmann, 2016*). Three species, *F. xylarioides*, *F. camptoceras* and *F. solani*, were the most virulent species causing Fusarium RR in sugar beet grown in Egypt (*Abo-Elnaga & Amein, 2011*; *Abd-El-Khaira, Abd-El-Fattahb & El-Nagdic, 2013*), whereas two species, *F. solani* and *F. culmorum*, were reported as the casual agents of the disease in the UK (*Jacobsen, 2006*). In the USA, *F. oxysporum* has been identified as a major cause of Fusarium RR in sugar beet (*Harveson & Rush, 1997, 1998*; *Jacobsen, 2006*; *Christ & Varrelmann, 2011*), as well as in Moldova and in the south-eastern part of Kazakhstan (*Maui, 2002*; *Lupashku & Mereniuc, 2010*; *Urazaliev et al., 2013*).

Infections in the roots of sugar beet plants in the field are known to be caused by a complex of several phytopathogens in different countries (*Jacobsen, 2006*; *Liebe et al., 2016*), including Kazakhstan, where three major strains of *F. oxysporum* Sch., No. 5, 50 and 150, were isolated (*Maui, 2002*; *Mombekova et al., 2013*). Therefore, it is important to define *F. oxysporum* strains, which can be identified in field trials. To this end, artificial inoculations of young plants with a single *Fusarium* strain in the laboratory can be used to accurately estimate virulence as well as the resistance of sugar beet genotypes to the fungus (*Harveson & Rush, 1998*). Finally, expression analysis of candidate genes must be carried out and compared with the symptoms of natural fungal infection in field tests and with symptoms in laboratory tests using inoculations of a single *Fusarium* strain. This can help to identify and verify possible candidate genes for Fusarium RR resistance and shed light on the different reactions of sugar beet plants to *Fusarium* infection (*Larson, Hill & Nuñez, 2007*).

*Fusarium oxysporum* is part of a diverse fungal genus containing many species that include pathogenic or non-pathogenic, highly virulent or avirulent forms causing RR, wilt and other disease symptoms in various plant species (*Mirkova & Karadjova, 1994*; *Harveson & Rush, 1997*; *Christ & Varrelmann, 2011*; *Webb, Covey & Hanson, 2012*; *Webb et al., 2013*; *Covey et al., 2014*). The metabolites produced by *F. oxysporum* are so toxic that they have been successfully used as a biocontrol agent in sunflower seeds and seedlings against parasitic plants of the genus *Orobanche* (*Amsellem et al., 2001*; *Dor et al., 2007*; *Fan et al., 2007*). Nevertheless, it was reported that mango and oil palm plants have the capacity to suppress or reduce pathogen attack by inducing the synthesis of chitinase (*Ebrahim, Usha & Singh, 2011*; *Rusli, Idris & Cooper, 2015*).

The activity of chitinase enzymes (EC 3.2.1.14) is correlated with pathogenic infection and, therefore, may play an important role in the plant defence mechanisms within root or leaf cells against various phytopathogens, like *F. oxysporum*. For example, in sugar beet, chitinases were shown to be highly up-regulated in plants inoculated with *F. oxysporum* strain F-19, a fungus associated with the root disease Fusarium yellows (*Larson, Hill & Nuñez, 2007*). Similar results showing strongly increased chitinase production were reported in sugar beet leaves infected by *Cercospora beticola* (*Nielsen et al., 1993*, *1994*), vascular wilt due to root infection by *F. oxysporum* in oil palm, *Elaeis guineensis* (*Rusli, Idris & Cooper, 2015*), and resistance to floral malformation in mango, *Mangifera indica* L. (*Ebrahim, Usha & Singh, 2011*). Chitinases catalyze the hydrolysis of chitin, which is present in the cell walls of fungi and insects but is not found in plant cells (*Collinge et al., 1993*; *Nagpure, Choudhary & Gupta, 2014*). Therefore, chitinases may be associated with tolerance to Fusarium RR in other crops including sugar beet plant species.

Chitinases fall into five classes, some of which are specific to plants only and others that are also present in bacteria and fungi (*Roopavathi, Vigneshwari & Jayapradha, 2015*), and are produced by plants in response to infection by either a single or particular group of pathogens (*Roopavathi, Vigneshwari & Jayapradha, 2015*). Sugar beet is a plant species capable of producing chitinase in association with a response to pathogenic fungi (*Nielsen et al., 1993*, *1994*; *Larson, Hill & Nuñez, 2007*). Chitinases have received significant attention due to their effectiveness in a wide range of applications, including their use as a biocontrol agent against plant-pathogenic fungi (*Nagpure, Choudhary & Gupta, 2014*).

Two isoforms of acid chitinase (SE1 and SE2) were identified in leaves of sugar beet in response to infection by the leaf-spot fungus *Cercospora beticola,* but only one of the isoforms (SE2) showed exochitinase activity and could effectively hydrolyse chito-oligosaccharides (*Nielsen et al., 1993*). SE2 shows most similarity to a Class III chitinase in *Arabidopsis*, tobacco and cucumber. The sugar beet *SE2* gene (designated as *BvSE2* = *B. vulgaris SE2*) showed a much higher level of expression after infection by *Cercospora* in tolerant compared to susceptible cultivars (*Nielsen et al., 1993*). Application of a cell-wall protein solution extracted from the non-pathogenic oomycete *Pythium oligandrum* isolate onto sugar beet leaves initiated a strong defensive reaction. This involved the significant increase of *BvSE2* gene expression, with maximal value at 4 h after inoculation (*Takenaka & Tamagake, 2009*). However, no chitin is present in oomycetes, so *BvSE2* may be induced as part of a co-ordinated response to other pathogenesis-related proteins (*Collinge et al., 1993*).

Similar to SE1 and SE2, two isoforms of another acid chitinase (SP1 and SP2) were also found and described in leaves of sugar beet infected by *C. beticola* (*Nielsen et al., 1994*). Only one glycosylated isoform, SP2, was characterized as a Class IV chitinase, similar to *SP2* genes found in rapeseed, bean and maize (*Collinge et al., 1993*). Expression of the *BvSP2* (= *B. vulgaris SP2*) gene was very high in infected leaves of sugar beet plants, but BvSP2 protein accumulation was limited to the area immediately surrounding the sites of infection (*Nielsen et al., 1994*). Sugar beet chitinase IV gene *BvSP2* when expressed in
transgenic birch (*Betula pendula*) inhibited growth of the insect larvae of *Orgyia antiqua* L., Lymantriidae (*Vihervuori et al., 2013*) but had no adverse effects on mammalian herbivores like roe deer (*Capreolus capreolus* L.) (*Vihervuori et al., 2012*).

Molecular markers, including single nucleotide polymorphism (SNP), are currently employed as routine method for the molecular genetic analysis of candidate genes. Amplifluor (Amplification with fluorescence) SNP analysis is a novel technology for high-throughput, accurate SNP genotyping that is based on a similar platform to Kompetitive allele specific PCR markers (*He, Holme & Anthony, 2014*). Amplifluor-like SNP markers have a simple design, employing a mixture of two universal probes carrying fluorescent labels and a quencher, and non-labeled gene-specific primers (*Giancola et al., 2006*; *Shavrukov et al., 2016*; *Jatayev et al., 2017*). Based on our recent study in bread wheat (*Shavrukov et al., 2016*), we extended our research to sugar beet via the development of primers targeting SNPs in an effort to identify candidate genes for RR resistance.

The aims of this study were: (1) to evaluate a set of sugar beet breeding lines and accessions for Fusarium RR resistance in comparative tests between artificial inoculation with *F. oxysporum* Sch. in the laboratory and natural infections of an unidentified pathogen(s) in field trials; (2) to identify SNPs for the design and scoring of SNP Amplifluor-like markers specific to the two chitinase genes *BvSE2* and *BvSP2*; and (3) to analyze gene expressions in response to inoculation with *F. oxysporum*, strain No. 5, and infection in field trials.

## MATERIALS AND METHODS

### Plant material

The 22 sugar beet breeding lines (accessions) used in this study are listed in Table S1 with their corresponding origins. All breeding material is self-incompatible, with population-based selection and propagation employed for their production. Seeds were supplied by the Taldykorgan branch of the Kazakh Research Institute of Agriculture and Plant Growing, Taldykorgan, Almaty district, Kazakhstan. Seeds of KWS2320 were kindly provided by KWS SAAT SE, Einbeck, Germany. The genome of KWS2320 as the Reference line has been completely sequenced (*Dohm et al., 2014*) and is publicly available on the NCBI database (accession number. PRJNA74567; www.ncbi.nlm.nih.gov).

### Artificial inoculation with *F. oxysporum* fungal suspension in the laboratory and resistance scoring of infected plants

For the laboratory test, sugar beet seedlings were grown in 10-cm diameter pots filled with field-collected, autoclaved soil (120 °C, 20 min treatment) in a greenhouse at 28/22 °C with 16/8 h day/night to the 3–4 leaf growth stage. Fungal inoculation was carried out as described by *Walker (1969)* with the following steps. Cuts were made with a sterile scalpel measuring one cm in length, penetrating approximately 1/3 of the width in the top part of the root in the pot-grown plants. *F. oxysporum* Sch., strain No. 5, was isolated earlier from infected roots in a previous field trail among two other major strains No. 50 and 150

(*Maui, 2002*). The isolates of *F. oxysporum* were inoculated onto the surface of Czapek media and grown until they were undergoing rapid growth, as described earlier (*Thom & Church, 1926*). All types of conidia were collected by scalpel and mixed into 1 ml of sterile water. One drop (50 μl) of the resulting suspension was added into the incision and allowed to absorb into the root tissue. The inoculation was repeated 10 days later in the same razor cut and in the same plants. Fusarium RR symptoms were assessed after one month (on leaf blades and petioles) and two months (in roots) following the date of the first inoculation. In laboratory tests, 10 plants were scored in each accession.

The field trials were conducted in the Taldykorgan, Almaty district, on south-eastern part of Kazakhstan during two years (2016 and 2017). Plants of each breeding line were grown in a 16 m row, with four rows in each plot, 60 cm between rows, 18–20 cm between plants, and a planting density of 5–6 plants within 1 m of a row. A total of four replicates of plots in each breeding lines were completely randomized. Plants in the field trial received no artificial inoculations. A total of 10 plants growing in the same general area were randomly selected for scoring in each replicate in the field trial; scoring was repeated for four replicate plots in each accession. The total number of scored biological replicates (plants) in field was 40 for each breeding line.

An identical five-point scoring system was applied in both the laboratory tests and the field trial, as follows: **0**—Highly resistant (disease development 0–15% of plants); **1**—Stable (disease development 16–30% of plants); **2**—Average resistance (disease development 31–50% of plants); **3**—Susceptible (disease development 51–70% of plants); **4**—Highly susceptible (disease development 71–100% of plants) (*Khovanskaya et al., 1985*; *Urazaliev et al., 2013*). Each score was based on two assessments (leaf blades/petioles and roots) in the same inoculated plants in the laboratory test and in plants without inoculation in the field trial. If scores in both assessments were identical, it was recorded as a common result, but both scoring results were recorded in the form of a range, if the two assessment scores differed. Non-inoculated plants in the laboratory test were used as controls with a **0** score.

## DNA extraction, bioinformatics and sequencing for SNP identification

DNA was extracted from young plants at the 5–6 leaf stage, using the CTAB method (*Dellaporta, Wood & Hicks, 1983*). The concentration of DNA was measured using a NanoDrop (Thermo Fisher Scientific, Waltham, MA, USA) and was then adjusted with sterile water to 10 ng/μl.

DNA sequences were obtained from the annotated line KWS2320 on the NCBI database (www.ncbi.nlm.nih.gov); *BvSE2* from LOC104888158 and *BvSP2* from LOC104888888. Primers were designed to cover entire genes, based on the annotated sequences. Sequences of *BvSE2* and *BvSP2*, as well as primers for sequencing and gene-specific SNP markers KIZ4 and KIZ3, as well as the positions of SNPs identified in this study, are provided in Figs. S1 and S2, respectively. Amplified PCR products for *BvSE2* and *BvSP2* genes (1,076 and 1,609 bp, respectively), were purified using a PCR Purification kit (Qiagen, Melbourne, Australia) and Sanger sequenced from both 5′- and

3′-ends using a service in the Australian Genome Research Facility, Adelaide, Australia. The comparison of *BvSE2* and *BvSP2* sequences in breeding lines from Kazakhstan and KWS2320 was used to test for the presence of SNPs. Two independent biological replicates were used to verify the accuracy and stability of the sequence. The SNP positions identified were then used for primer design and SNP Amplifluor-like analysis.

## SNP Amplifluor-like genotyping assay

Amplifluor-like SNP analysis, as described earlier (*Shavrukov et al., 2016*; *Jatayev et al., 2017*), was carried out using a QuantStudio-7 Real-Time PCR Cycler (Thermo Fisher Scientific, Waltham, MA, USA). The PCR cocktail in each well contained 2xMaster-Mix with the following reagents in their final concentrations: 1.8 mM $MgCl_2$, 0.25 µM each fluorescent label probe, 0.2 mM each of dNTPs, 0.15 µM of each forward primer, 0.78 µM of reverse primer, 1× PCR Buffer and 0.5 units of Taq DNA polymerase (GenLab, Astana, Kazakhstan). Half of the total PCR volume (5 µl out of 10 µl) was genomic DNA, adjusted for 10 ng/µl, and 5.0 µl or 2.5 µl of each DNA sample was added in 96- or 384-well microplates, respectively. One µl of 1:100 diluted ROX was added as a passive reference label to the Master-Mix according to the manufacturer's protocol for the PCR Cycler.

PCR cycling followed a program adjusted from those published earlier (*Shavrukov et al., 2016*; *Jatayev et al., 2017*), and included initial denaturation at 95 °C for 1 min; 26–28 cycles of 95 °C for 30 s, 55 °C for 30 s and 72 °C for 50 s; and final extension at 72 °C for 5 min. Genotyping with SNP calling was determined automatically by software accompanying the instrument, but each SNP result was checked manually using amplification curves and allele discrimination. Genotyping experiments were repeated twice (two technical replicates) over different days, where technical replicates confirmed the confidence of SNP calls.

Genotyping of eight individual plants in each breeding line was scored based on the majority of the identified alleles in SNP analysis for both studied genes, $BvSP2\text{-}a_1a_1$; $\text{-}a_1b_1$; $\text{-}b_1b_1$; and $BvSE2\text{-}a_2a_2$; $\text{-}a_2b_2$; $b_2b_2$. The same plants were used for phenotyping scores, where all studied breeding material was highly heterogeneous for both *BvSE2* and *BvSP2*. Reference line KWS2320 was homozygous for both genes, $BvSP2\text{-}a_1a_1$ and $BvSE2\text{-}a_2a_2$.

## RNA extraction, cDNA construction and qPCR analysis for gene expression

Plants were grown in the laboratory until 3–4 leaf growth stage, as described above. All leaves were collected from each plant immediately prior inoculation with *F. oxysporum* Sch., strain No. 5 (Day 0, used as Control), with following similar samplings of other plants from the same pots, 14 and 18 days after the inoculation, based on our preliminary tests. In the field trial, two spots were designated as non-infected (Controls, C), where plants had no symptoms of RR and others were designated as infected (I) areas where they showed severe RR symptoms after one month of growth. Two biological replicates for each treatment and control were used. RNA was extracted from leaf samples of individual

plants both in the laboratory and in the field trial using TRIsol-like reagents, as published earlier (*Shavrukov et al., 2013*). RNA quality was checked by agarose gel electrophoresis with following treatments for 15 min, at room temperature, with 1 μl of DNase I (Thermo Fisher Scientific , Waltham, MA, USA) to remove any traces of DNA. cDNA was constructed from 2 μg of DNase-treated RNA using oligo-dT$_{(18)}$ and Reverse Transcriptase kit, M-MuLV-RH (Biolabmix, Novosibirsk, Russia; http://www.biolabmix.ru) following the manufacturer's instructions.

Non-diluted cDNA samples were used for qPCR analyses in a Real-Time PCR system, Model CFX96 (BioRad, Gladesville, NSW, Australia) at Flinders University, Australia. The total volume of 10 μl qPCR reactions included 5 μl of 2× KAPA SYBR FAST (KAPA Biosystems, Wilmington, MA, USA), 4 μl of cDNA, and 1 μl of mixed forward and reverse gene-specific primers (3 μM of each). Expression data for the target genes, *BvSP2* and *BvSE2*, were normalized using the reference gene, Glutamine synthetase (LOC104883503), as suggested and used for sugar beet (*Taski-Ajdukovic et al., 2012*) and repeated twice (two technical replicates). It was checked with infected and control plants as a suitable reference gene prior to the analysis. The Relative standard curve method is based on the ABI Guide to performing relative quantitation of gene expression using real-time quantitative PCR (http://www3.appliedbiosystems.com/cms/groups/mcb_support/documents/generaldocuments/cms_042380.pdf), where serial dilutions were used for each target and reference gene individually (*Soole & Smith, 2015*). Based on linear calibration of template cDNA dilution factor and Cq value, the threshold cycle values were determined. The coefficients of determination, $R^2$, were greater than 0.995 in all studied target and reference genes. The efficiencies of all qPCR primers were calculated based on slope of the corresponding calibration line, $E = 10^{slope}$ (*Borges, Tsai & Caldas, 2012*), and they were in the suitable range, 1.8–2.0. Specificities of target and reference genes were confirmed by single distinct peaks on a melting curve and a single band of the expected size in 2% agarose gel electrophoresis. Details of primer sequences and product sizes are present in Table S2.

## Statistical analysis

The non-parametric Mann–Whitney *U*-test was applied for ranking scores in pair comparisons, as suggested for plant disease severity (*Shah & Madden, 2004*), and the online calculator was used at http://www.socscistatistics.com/tests/mannwhitney. The Mann–Whitney *U*-test in this case operates with a Nominal variable or Categorical (ordinal) variable data (*Harris et al., 2008*), and also with a small number of analyses in each studied accession (*Fagerland, 2012*), making it the most suitable statistical treatment for ranking score comparisons in pairs. The paired Sign test was used to analyze if two sets of scores were correlated with each other using the online calculator http://www.socscistatistics.com/tests/signtest/Default.aspx. For gene expression analysis, average and standard errors were calculated using standard Excel software. Probabilities for significance, $P < 0.05$ and $P < 0.01$, were calculated using Student's *t*-test.

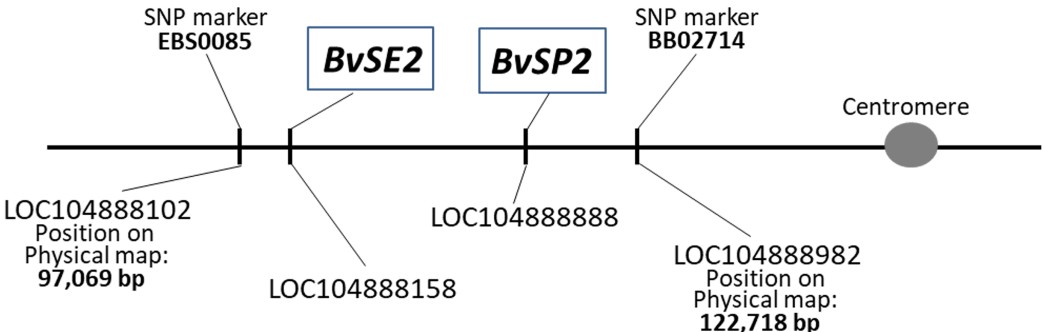

## Chromosome 3S

**Figure 1 Positions of *BvSE2* and *BvSP2* genes, encoding chitinases, Classes III and IV, respectively, on the physical map of chromosome 3S in sugar beet.** SNP markers were published by *Holtgräwe et al. (2014)*, and LOC sequences were identified from the NCBI database.

## RESULTS

### Structure of the chitinase genes *BvSE2* and *BvSP2,* and their localisation on chromosome 3S

Sequences of *BvSE2* and *BvSP2* were annotated using sequence obtained from the sugar beet genome database (NCBI). The full sequence of sugar beet *BvSE2* chitinase, Class III was identical in all annotated accessions (S66038, AAB28479, XM_010673039 and LOC104888158) including accession number 567433940 for the doubled haploid (DH) line KWS2320 using a Whole genome shotgun sequencing approach (WGSGS). The ORF was 882 base pairs (bp) with 293 predicted amino acids (aa) making up the corresponding protein BvSE2 (accession XP_010671341). Similarly, sugar beet annotated accessions, L25826, AAA32916, XM_010673997 and LOC104888888 with accession 730225012 for WGSGS, were identified for the sequence of *BvSP2* chitinase, Class IV. In the comparison of the two chitinases, both ORF of the *BvSP2* gene and the BvSP2 protein sequence were shorter, comprising 867 bp and 288 aa, respectively (XP_010671341). No SNP or other polymorphism was found in the databases for either *BvSE2* or *BvSP2* genes.

The chromosome locations of *BvSE2* and *BvSP2* on sugar beet chromosome 3 were indicated in the annotated LOC accessions. The positions of these genes on the physical map of chromosome 3S were identified using published information with flanking SNP markers, EBS0085 and BB02714 (*Holtgräwe et al., 2014*). Using WGSGS, the genes were located in the interval between 97,069 and 122,718 bp on the short arm of chromosome 3; the position of *BvSE2* was determined as distal and close to SNP marker EBS0085, while *BvSP2* was proximal and located slightly further from the SNP marker BB02714 (Fig. 1).

### Sequence analysis revealed SNPs in *BvSE2* and *BvSP2*

Two sugar beet breeding lines (2217 and 2263), with different germplasm origin (Table S1), were selected for sequencing. Two biological replicates were included for each

accession. The sequenced genetic regions in the *BvSE2* and *BvSP2* genes comprised 1,076 and 1,609 bp, respectively. The fragments encompass both 5′- and 3′-UTR, and cover entire genes, including introns. Sequencing of *BvSE2* and *BvSP2* fragments from both ends of the DNA fragments revealed the full sequence identity when compared to KWS2320. A single SNP was identified in the *BvSE2* and *BvSP2* genes, with substitutions $[W = A/T]$ and $[Y = C/T]$ at positions 377 and 345 from the Start-codon, in *BvSE2* and *BvSP2*, respectively. Both identified SNPs were heterozygotes in the two biological replicates studied. This means that one allele at each SNP position was identical to the sequence in KWS2320 but the other differed. It is important to note that the SNP $[W = A/T]$ corresponded to a substitution of $N_{126}$ with $I_{126}$ in the predicted amino acid sequence of the BvSE2 protein, but the SNP $[Y = C/T]$ of $R_{115}$ was synonymous.

### Phenotyping of the response to *F. oxysporum* Sch. inoculation, strain No. 5 and genotyping with an SNP Amplifluor-like marker for *BvSP2* chitinase gene, Class IV

A total of 10 individual plants for each of 22 accessions plus KWS2320 were examined for response to *F. oxysporum* Sch., strain No. 5, as described in the Materials and Methods. Laboratory phenotype scores assessing the apparent resistance levels one and two months after inoculation were consistent. An overview of scoring results is presented in Table 1. Breeding lines 1017, 2282 and 2296 showed the highest resistance (score of 0–1) to *F. oxysporum*, strain No. 5, identical to those of KWS2320. More than half of all studied accessions were classified as resistant, with scores ranging between 1 and 2. Three lines had an equal number of resistant and susceptible plants. Two lines (2125 and 2210) after inoculation with *F. oxysporum* strain No. 5, showed the strongest symptoms of susceptibility to the phytopathogen (Table 1A).

Genotyping results were based on self-designed Amplifluor-like SNP markers, and amplification with either FAM or VIC produced fluorescence signals linked with the corresponding SNP. This quantitative fluorescence was determined by the qPCR instrument and used for allele discriminations with automatic SNP calls among individual plants in different breeding lines. Examples of allele discrimination for the SNP in *BvSP2* and *BvSE2* are presented in Fig. 2. The genotypes formed clearly separate two groups with homozygotes and relatively small portion of heterozygote SNPs. All genotypes of KWS2320 plants fell into the homozygote classes, *BvSP2-a₁a₁* and *BvSE2-a₂a₂*.

The genotyping results of plants used for Fusarium RR phenotyping, when tested with the SNP Amplifluor-like marker KIZ3 for the *BvSP2* gene, revealed similar but not identical results (Table 1A). Only KWS2320 had identical genotyping scores for all studied plants, while the majority of the genotypes, *BvSP2-a₁a₁*; *-a₁b₁*; and *-b₁b₁*, were estimated in the other breeding lines. Genotyping results showed a high level of association (20 out of 22 lines in total) with the phenotypic estimation of the resistance or susceptibility to *F. oxysporum*, strain No. 5 ($P < 0.01$, Sign test). Two cases of non-opposed mismatches for *BvSP2* gene were found between phenotyping and genotyping scores

**Table 1 Comparison of disease scores for *Fusarium oxysporum* Sch.**

| Breeding line ID | A | | B | |
|---|---|---|---|---|
| | Laboratory score | Genotyping of *BvSP2* | Field score | Genotyping of *BvSE2* |
| 1002 | **1**\*, R | $a_1a_1$ | **2–3**\*, S | $a_2b_2$ |
| 1005 | 1–2, R | $a_1a_1$ | 2, In | $a_2b_2$ |
| 1017 | 0–1, R | $a_1a_1$ | 1–2, R | $a_2b_2$ |
| 1042 | 2, In | $a_1a_1$ | 2–3, S | $b_2b_2$ |
| 1082 | 1–2, R | $a_1a_1$ | 1, R | $a_2a_2$ |
| 2115 | 1–2, R | $a_1a_1$ | 1, R | $a_2a_2$ |
| 2125 | 2–3, S | $b_1b_1$ | 1–2, R | $a_2a_2$ |
| 2154 | 2, In | $a_1a_1$ | 2–3, S | $b_2b_2$ |
| 2172 | 1–2, R | $a_1a_1$ | 2, In | $a_2b_2$ |
| 2182 | 1, R | $a_1a_1$ | 0–1, R | $a_2a_2$ |
| 2190 | 1–2, R | $a_1a_1$ | 1–2, R | $a_2a_2$ |
| 2210 | **2–3**\*, S | $b_1b_1$ | **0–1**\*, R | $a_2a_2$ |
| 2217 | 2, In | $a_1b_1$ | 1–2, R | $a_2b_2$ |
| 2236 | **1–2**\*, R | $a_1a_1$ | **4**\*, S | $b_2b_2$ |
| 2261 | 1–2, R | $a_1a_1$ | 0–1, R | $a_2a_2$ |
| 2262 | 1–2, R | $a_1a_1$ | 2–3, S | $b_2b_2$ |
| 2263 | 2, In | $a_1b_1$ | 2, In | $a_2b_2$ |
| 2281 | 1–2, R | $a_1a_1$ | 1–2, R | $a_2a_2$ |
| 2282 | 0–1, R | $a_1a_1$ | 1, R | $a_2a_2$ |
| 2286 | 1–2, R | $a_1a_1$ | 1–2, R | $a_2a_2$ |
| 2296 | 0–1, R | $a_1a_1$ | 1–2, R | $a_2b_2$ |
| 2300 | 1, R | $a_1a_1$ | 1–2, R | $a_2a_2$ |
| KWS2320 | 0–1, R | $a_1a_1$ | 0–1, R | $a_2a_2$ |

**Notes:**

Comparison of disease scores for *Fusarium oxysporum* Sch., isolate No. 5 symptoms in sugar beet breeding lines, showing resistant, **R** (0 or 1), intermediate, **In** (2) and susceptible, **S** (3 or 4) scores, identified after inoculation in the laboratory (A) and growth in a field trial (B). Genotyping of *BvSP2* and *BvSE2* was made according to the majority of identified alleles, *BvSP2*-$a_1a_1$; -$a_1b_1$; -$b_1b_1$; and *BvSE2*-$a_2a_2$; -$a_2b_2$; $b_2b_2$. Mixed phenotyping scores of 1–2 or 2–3 in the laboratory and field trial were designated as resistant and susceptible genotypes, respectively. The number of plants in the laboratory test and field trial were $n = 10$ and $n = 40$, respectively, while ($n = 8$) plants were used for genotyping. All experiments were conducted twice. Significant differences between pairwise comparisons according to the Mann–Whitney *U*-test ($P < 0.05$) are shown in bold and marked with an asterisk (\*). Other data showed no significant differences in pairwise comparisons using the same *U*-test.

(in lines 1042 and 2154) with an intermediate phenotyping score (2) and genotyping of *BvSP2*-$a_1a_1$ alleles (Table 1A).

## Comparison of inoculation and field trial scores with genotyping using an SNP Amplifluor-like marker for the *BvSE2* chitinase gene, Class III

A comparison of disease severity ratings to Fusarium RR in sugar beet accessions using inoculation with *F. oxysporum* isolate, strain No. 5, and to RR in plants grown in a field trial is presented in Tables 1A and 1B. Most of the breeding lines (16 in the laboratory test and 14 in the field trial) out of 22 studied were resistant to RR. However, a number of accessions were classified as susceptible (score ratings 3 or 4) and differed in the

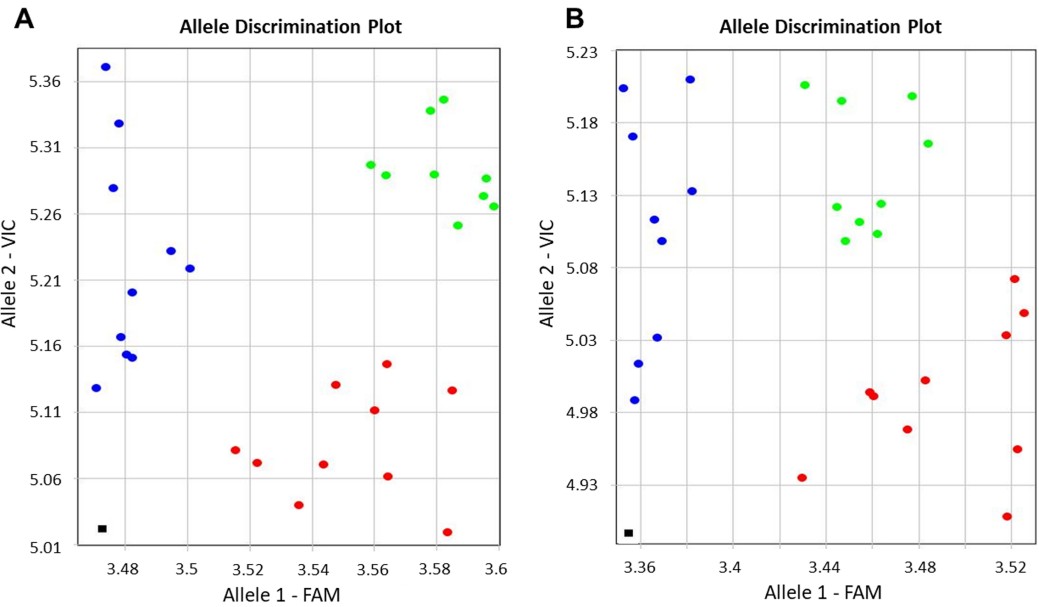

**Figure 2 Allelic discrimination of SNP Amplifluor-like markers KIZ3 (A) and KIZ4 (B), designed for the *BvSP2* and *BvSE2* genes encoding chitinases, Class IV and Class III, respectively, among sugar beet breeding lines.** Red and blue dots indicate automatic SNP calls for homozygotes in Allele 1 (*BvSP2-$a_1a_1$*; *BvSE2-$a_2a_2$*) and Allele 2 (*BvSP2-$b_1b$*; *BvSE2-$b_2b_2$*), respectively, while green dots indicate heterozygotes (*BvSP2-$a_1b_1$*; *BvSE2-$a_2b_2$*) or mixed genotypes. Black squares show NTC (No Template Control). *X*- and *Y*-axes show Relative amplification units, ΔRn, for FAM and VIC fluorescence signals, respectively, as determined by the qPCR instrument.

phenotyping score categories, accounting for two and five lines in the laboratory test and field trial, respectively.

Results of the comparison between scores for the laboratory (Table 1A) and field tests (Table 1B) revealed that more than half (12 out of 22) of the studied breeding lines showed identical or very similar score results. Seven lines had overlapping RR scores, with a mostly intermediate phenotype between resistant and susceptible or a mixture of both. This indicates a high degree of similarity between symptom scores of RR in the two different tests. Finally, three conflicting cases (Breeding lines 1002, 2210 and 2236) were found with opposing RR scores in the laboratory test and the field trial (Table 1, indicated in Bold).

Genotyping results using the SNP Amplifluor-like marker KIZ4 for *BvSE2* gene, *BvSE2-$a_2a_2$*; *-$a_2b_2$*; and *$b_2b_2$*, showed a strong association with the RR scores in plants grown in the field trial (Table 1B). Four out of five breeding lines susceptible to RR in the field trial (1042, 2154, 2236 and 2262) had a perfect match with the genotyping results. The remaining accessions susceptible to RR (1002 for field trial, and 2172 and 2263 for *BvSE2-$a_2b_2$* genotypes) showed intermediate scores in both the field trial and genotyping. Eleven breeding lines were classified as recording very similar RR scores; all other lines had overlapping RR scores. No conflicting results with opposing RR scores for phenotyping in the field trial and genotyping with SNP marker for *BvSE2* were found. KWS2023 was resistant to RR in both laboratory inoculation and in the field trial (Table 1).

Based on results presented in Table 1, three breeding lines with contrasting reactions to inoculation with *F. oxysporum*, strain 5 in the laboratory test and RR score in field trials, were selected for gene expression analysis. Line 2182 was identified as the most resistant in both tests; line 2210 was sensitive to strain 5 inoculation but resistant to RR in field trials; and, in contrast, line 2236 was resistant to strain 5 inoculation but sensitive to RR in field trials. There was no identified breeding line sensitive in both laboratory and field tests. KWS was used as comprehensive reference line.

## Gene expression using qPCR

Expression analysis of *BvSP2* and *BvSE2* genes after inoculation with *F. oxysporum*, strain 5, showed that three selected lines (2182, 2236 and KWS2320) had very high expression on Day 14 after inoculation, which returned to levels similar to Controls at Day 18. In contrast, in RR sensitive breeding line 2210, the expression of *BvSP2* was 4–7.5-fold smaller at Day 14 compared to three RR resistance lines but it was 200-fold higher on Day 18 (Fig. 3A). In leaf samples from the infected field trial, only accession 2236 showed significantly increased expression of *BvSE2* compared to other samples from infected field trials as well as to all Controls (Fig. 3B).

## DISCUSSION

Reduced yields and crop quality due to RR in sugar beet can be resolved through the introduction of resistant genotypes (*Jacobsen, 2006*; *Christ & Varrelmann, 2011*), a goal that is especially important for the region of Kazakhstan. Genetic polymorphism for resistance and susceptibility to *Fusarium* species causing Fusarium RR has been reported for infected Egyptian sugar beet cultivars in field trials (*Abd-El-Khaira, Abd-El-Fattahb & El-Nagdic, 2013*), and our own results presented herein provide similar findings for genetic diversity among studied sugar beet germplasms in Kazakhstan. In our experiments, only three breeding lines, 1017, 2282 and 2296, remained free of RR symptoms in the laboratory test, and three other breeding lines, 2182, 2210 and 2261, recorded no symptoms in the field trial. Most breeding lines segregated for RR resistance in the laboratory test, field trial or in both. This observation can be directly related to the high level of heterogeneity resulting from general self-incompatibility in the species *B. vulgaris* L. (*Kalia, 2005*), which is typical in sugar beet material derived from population-based selection (*Archimowitsch, 1956*). Nevertheless, the strong association (Sign test, $P < 0.01$) was found in our study between Fusarium RR in laboratory and SNP genotyping for *BvSP2* from one side (Table 1A), and between RR symptoms in field trials and SNP genotyping for *BvSE2* from another side (Table 1B). This can indicate that both studied chitinase genes, *BvSP2* and *BvSE2*, are very likely differentially involved in the reaction of plants to infection with *Fusarium* species. All mismatching cases between genotyping and phenotyping in Table 1 are "overlapped" with intermediate scores, and they can be explained by the probable high level of heterogeneity in the studied breeding lines.

In Moldova and Southern Kazakhstan, *F. oxysporum* was previously determined (*Lupashku & Mereniuc, 2010*; *Urazaliev et al., 2013*) to be a cause of Fusarium RR in sugar beet, where three major strains of *F. oxysporum* Sch., No. 5, 50 and 150, were isolated

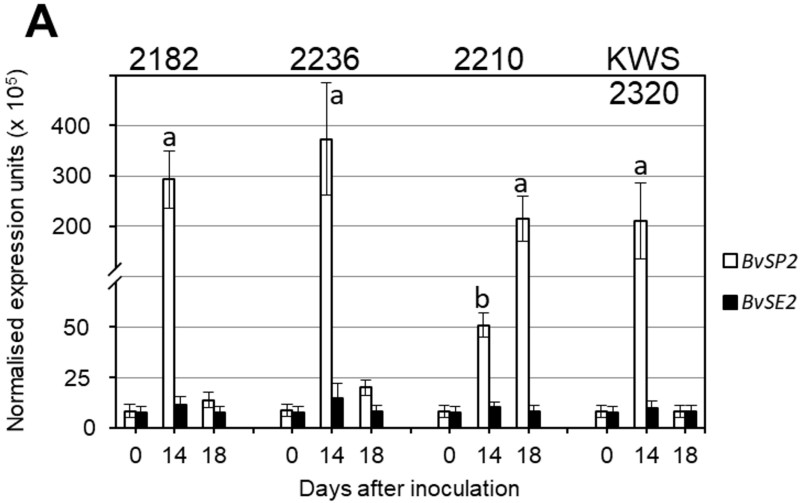

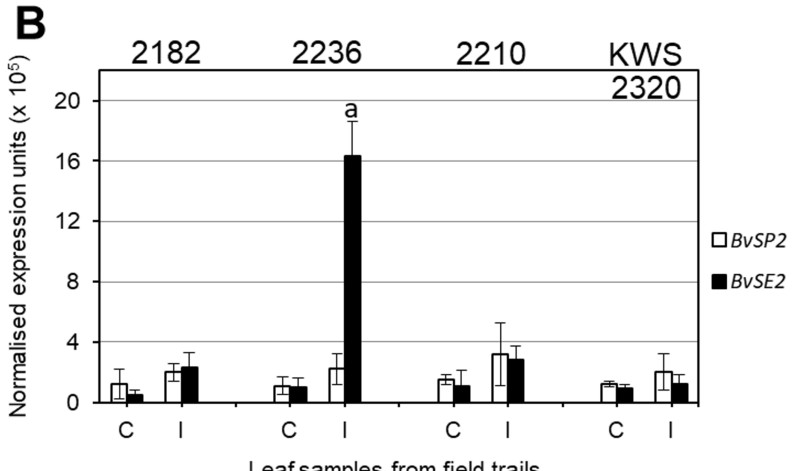

**Figure 3 Quantitative expression analyses (qPCR) of *BvSP2* and *BvSE2* genes (clear and black bars, respectively) in leaves of sugar beet plants.** (A) Plants at the 3–4 leaf stage were inoculated by *Fusarium oxysporum* Sch., isolate No. 5, and indicated Day 0, as Controls, for comparison with Days 14 and 18 since inoculations. (B) Mature plants from non-infected Controls (C), and Infected (I) parts of field trails. Names of the four selected sugar beet lines are shown on the top of each figure panel. Expression data were normalised using the reference gene, Glutamine synthetase (LOC104883503). Bars show the average for two biological and two technical replicates in each measurement ± Standard Error. Different letters indicate significant differences with other samples at the same time-points and Controls ($P < 0.01$ for (A) and $P < 0.05$ for (B)).

(*Maui, 2002*). Nevertheless, more precise taxonomic and molecular analyses are required to reliably distinguish *F. oxysporum* from other reported forms of the fungus in sugar beet (*Harveson & Rush, 1998*; *Jacobsen, 2006*; *Christ & Varrelmann, 2011*).

The fungal strains or species involved in causing Fusarium RR can vary considerably depending on geographic location, soil conditions, planting time, crop rotations in the field, herbicide applications and other factors (*Jacobsen, 2006*; *Lupashku & Mereniuc, 2010*). This means that cultivars or hybrids resistant to Fusarium RR in one area can be more or less susceptible in other areas. This may have contributed to the findings

presented in the current study, where three breeding lines (1002, 2210 and 2236) recorded conflicting RR scores in the field trial versus the inoculation by *F. oxysporum*, strain No. 5, in the laboratory. This indicates the presence of an additional factor influencing RR scores, very likely strains No. 50 or 150 of *F. oxysporum*, found earlier in the same field trial (*Maui, 2002*). Selected individuals from these accessions showing evidence of apparent resistance to Fusarium RR pathogens in one test and susceptibility in another should be carefully propagated and their seeds stored in GeneBank.

Of greater interest to sugar beet breeders are obviously genotypes showing symptoms of resistance to all (or at least to several) Fusarium RR pathogens. Based on the current study, we can identify only two breeding lines, 2181 and 2282, which have shown resistance to both *F. oxysporum* strain No. 5 and to RR infection in the field, with scores ranging between 0 and 1. It is also important to point out the genetic significance of genotypes showing a very high susceptibility to the studied RR pathogen, but these genotypes might be unsuitable for the growing region of Kazakhstan.

Very high expression levels of *BvSP2* gene in leaves of sugar beet plants after inoculation strongly support the hypothesis that this candidate gene is responsive to *F. oxysporum*, strain 5 (Fig. 3A). It is shown very clearly, and with very high significance ($P < 0.01$), that four days earlier (Day 14), the expression is associated with better resistance to the RR inoculation. This means that resistance to RR is very likely determined by a signaling system connecting with a defence mechanism, and this process is initiated by rapid and very strong expression of *BvSP2* in leaves (and perhaps other parts of plants) in response to inoculation by *F. oxysporum*, strain 5. In contrast, a delay in *BvSP2* expression response can slow down the defence mechanism and finally result in the sensitivity to RR, as seen in the breeding line 2210 (Fig. 3A).

Different time points were used in leaf samplings for analysis of *BvSP2* and *BvSE2* gene expression: 14 and 18 days after inoculation in the laboratory test and one month in the field trial test, respectively. Changes in *BvSP2* expression indicated a quicker response to the inoculation in resistant lines 2282, 2236 and KWS2320 (Fig. 3A) with maximum expression at Day 14. This type of expression profile was very similar to those for *BvSE2* after one month of growth in the field trial, where the significant changes of *BvSE2* gene expression were observed only in leaf samples of accession 2236 collected from infected field trials (Fig. 3B). The expression of *BvSE2* was very specific and occurred only in this single line, which displayed severe symptoms of RR in the field test. Three RR resistant lines 2182, 2210 and KWS2320 typically show earlier responses in *BvSE2* expression in response to RR infections in the field, but *BvSE2* expression levels returned back to initial levels after one month of plant growth in the infected field trials. However, this hypothesis is hard to assess in field conditions where the timing of natural infections is not controlled and can be verified only by inoculation tests in the laboratory with known strains of *F. oxysporum*.

It is important to point out that no significant changes were registered in the expression of *BvSP2* in the field test, which could indicate that strain 5 of *F. oxysporum* is not a dominant strain in field conditions or that perhaps the natural activity and aggressiveness of this strain in soil was much reduced compared to artificial inoculation in the laboratory test.

Therefore, we can speculate that the *BvSE2* gene may be responsive to strains No. 50 or 150 of *F. oxysporum*, which were discovered and described earlier in the same field trials (*Maui, 2002*).

At this stage, we can conclude that both candidate genes, *BvSP2* and *BvSE2*, demonstrated differential expression responses after artificial inoculation and natural infection in soil, which is likely due to the differing compositions of *F. oxysporum* strains causing RR. The final step in the research would be to study the signaling system and protein–protein interactions between the pathogen and host plant to complete the story on how *F. oxysporum* causes Fusarium RR and how resistance or susceptibility arises in sugar beet plants in our experiments. Nevertheless, results presented in the current study indicate that both chitinase genes, *BvSP2* and *BvSE2*, are strongly expressed in the plant in response to Fusarium RR. Interestingly, their roles appear to differ depending on which strain of *F. oxysporum* is present, or possibly in response to interactions with environmental factors or other pathogenic species; an observation that requires further investigation. This wider and more complex view of the mechanisms generating resistance to *F. oxysporum* isolate in sugar beet must be resolved in future studies.

## CONCLUSION

A strong association of two SNP markers for *BvSP2* and *BvSE2* with resistance to RR in sugar beet was found in our study. Very high *BvSP2* expression (100-fold compared to Controls) was observed in three RR resistant breeding lines (2182, 2236 and KWS2320) 14 days after inoculation with *F. oxysporum*, strain 5, and it was return to the control level on Day 18. RR sensitive breeding line 2210 showed a delay in mRNA level, reaching maximal expression of *BvSP2* 18 days after inoculation. The gene *BvSE2*, showed a strong expression level in leaf samples of one month-old plants from the infected field trial only in the breeding line 2236, which showed symptoms of RR, and this may be a response to other strains of *F. oxysporum*.

### Funding

This research was supported by Grant 4785/GF4, the Program 217 "Development of Science," Sub-program 102 "Grant Research Funding," from the Ministry of Education and Science, Kazakhstan. The funders had no role in study design, data collection and analysis, decision to publish, or preparation of the manuscript.

### Grant Disclosures

The following grant information was disclosed by the authors:
Ministry of Education and Science, Kazakhstan: Grant 4785/GF4, Program 217 Development of Science, Sub-program 102 Grant Research Funding.

### Competing Interests

The authors declare that they have no competing interests.

## Author Contributions

- Raushan Yerzhebayeva conceived and designed the experiments, analyzed the data, contributed reagents/materials/analysis tools, prepared figures and/or tables, authored or reviewed drafts of the paper, approved the final draft.
- Alfiya Abekova performed the experiments, analyzed the data, prepared figures and/or tables, approved the final draft.
- Kerimkul Konysbekov conceived and designed the experiments, performed the experiments, prepared figures and/or tables, approved the final draft.
- Sholpan Bastaubayeva contributed reagents/materials/analysis tools, approved the final draft.
- Aynur Kabdrakhmanova conceived and designed the experiments, performed the experiments, approved the final draft.
- Aiman Absattarova contributed reagents/materials/analysis tools, approved the final draft.
- Yuri Shavrukov analyzed the data, contributed reagents/materials/analysis tools, prepared figures and/or tables, authored or reviewed drafts of the paper, approved the final draft.

## Data Availability

The raw data are provided in the Supplemental Files.

## Supplemental Information

Supplemental information for this article can be found online at http://dx.doi.org/10.7717/peerj.5127#supplemental-information.

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
