# Peer review of "Two sugar beet chitinase genes, BvSP2 and BvSE2, analysed with SNP Amplifluor-like markers, are highly expressed after Fusarium root rot inoculations and field susceptibility trial"

_PeerJ, doi:10.7717/peerj.5127_

## Round 0.1 · original submission · Minor Revisions

There are really only technical remarks from both reviewers. Please take them into account.

Reviewer 1 ·

Basic reporting

The article is devoted to the urgent problem of search and identification of candidate genes participating in the reaction to biotic stress forms. As an object, the important food crop, sugar beet is used, and the pathogen is the fungus Fusarium, which causes a widespread disease of roots. It was previously shown that chitinase genes are one of the first to be included in response to the plant pathogens. The authors set a reasonable task to identify the association between certain isoforms of these genes and both resistant and fusarium-sensitive plant forms, and also to analyze the gene expression in response to specific fungal strains most common in Kazakhstan.
The article is nicely written in English, all terms are correctly used. It includes sufficient introduction with complete and clear review the scientific problem under study. However, some paragraphs can be rearranged. Since it starts with description of Fusarium pathogen, so paragraphs under lines 98 and 132 (which disrupt the description of chitinases) can be combined with the corresponding part of Introduction.
The structure of the article agrees with appropriate form. Figures are relevant to the content of the article and have sufficient resolution, except Fig.2 where numbers at X- and Y-axes look very small.
Minor remark. Line 334- amino acid substitution should be indicated for non-synonymous case.
More serious comment. One of the main tasks of this study is identification of genes and their markers suitable for the selection of plants most resistant to the pathogen. However, in my opinion, the part devoted to SNP-targeting of alleles looks separate from the analysis of expression and phenotyping. There should be some clarification, is there association between the SNPs found and resistance to Fusarium, or they are neutral? There is a phrase in conclusion: “A strong association with two SNP Amplifluor-like markers with BvSP2 and BvSE2 was found in our study”. But any other SNP may serve as a marker of these genes, but not a marker for resistant forms. You found rise of expression of these genes, but could you discriminate between different plant forms using your markers? Explain, please.

Experimental design

Original research is within Aims and Scope of the journal. Experimental design agrees well with the proposed tasks and the study contributes to resolve these tasks.

Validity of the findings

Data is robust, statistically sound and controlled. Conclusion is well stated, linked to original research question and limited to supporting results.

Additional comments

As a whole, the article is ready for publication after a minor editing will be done.

Reviewer 2 ·

Basic reporting

No comment

Experimental design

No comment

Validity of the findings

No comment

Additional comments

The manuscript represents the original research devoted to the determination of the role of BvSP2 and BvSE2 genes in resistance of sugar beet to Fusarium root rot. The association of sugar beet resistance with SNPs and expression of the genes has been shown. In general, the manuscript is well structured and written in a professional way, the references are sufficient and the figures are clear. The results were obtained with the use of representative sample sets and rigorous techniques and correctly interpreted. However, there are several issues that should be addressed:

1) Figure 3A – Why there are asterisks only over 2210 line? Hundred-fold expression increase in 2182, 2236, and KWS2320 lines is not statistically significant? Also, it is better to present Y-axis in logarithmic scale, it is impossible to understand what values are shown slight above 0.
2) Table 1 – Strange upper row – “A” and “B”, it can be concluded that genotyping for BvSP2 gene was performed only for laboratory-grown plants and that for BvSE2 gene – only for field-grown plants. Is it correct? Please, correct or explain. Why are “R”, “In”, and “S” abbreviations presented in columns 3 and 5 “Genotyping”? Please, correct or explain.
3) Lines 358-359 – “All genotypes of KWS2320 plants fell into the homozygote class a1a1 (Fig. 2).” I cannot find any mention on KWS2320 neither in Figure 2 nor in Figure 2 legend. Please, correct or explain.
4) Lines 296-297 – “Probabilities for significance, P<0.95 and P<0.99, were calculated using Student’s t-test.” This looks strange for me, what do you mean?
5) Line 279 – R2 value has the relation to correlation coefficient, but it is not correlation coefficient. Please, correct.
6) qPCR is used as synonym to real-time PCR, while qRT-PCR is used to abbreviate quantitative reverse transcription PCR. It should be corrected where applicable. Also, the correct name of Bio-Rad CFX96 device is “Real-Time PCR System” not “Real-Time qPCR System”. Please, correct.
7) Lines 335-336 – “the SNP [Y = C/T] was non-synonymous” The SNP is synonymous. Please, correct.
8) Line 261 – “ThermoFisher, Almaty, Kazakhstan”. Is it correct?
9) Lines 515-516 Strange conclusion on association of SNP markers with genes. Did you mean the association of genotype with the resistance to root rot?

I recommend to publish the manuscript after minor revision.

---

## Round 0.2 · accepted · Accept

The manuscript had minor remarks for the first version. Now both reviewers have no more comments. Thank you again for interesting materials.

# Reviewer 1 ·

Basic reporting

ok

Experimental design

ok

Validity of the findings

ok

Additional comments

I'm completely satisfied with the author's answer.

Reviewer 2 ·

Basic reporting

No comment

Experimental design

No comment

Validity of the findings

No comment

Additional comments

The authors have addressed all the comments. I recommend to accept the manuscript for publication in PeerJ.